# Influence of High-Pressure Torsion and Accumulative High-Pressure Torsion on Microstructure and Properties of Zr-Based Bulk Metallic Glass Vit105

**Dmitry Gunderov [1,2,]***[ID]**, Vasily Astanin [3]**[ID]**, Anna Churakova [1,2]**[ID]**, Vil Sitdikov [1]**[ID]**, Evgeniy Ubyivovk [4]**[ID]**, Akhmed Islamov [5] and Jing Tao Wang [6]**[ID]

[1]  Institute of Physics of Advanced Materials, Ufa State Aviation Technical University, 12 K. Marx Str., 450008 Ufa, Russia; churakovaa_a@mail.ru (A.C.); svil@ugatu.su (V.S.)
[2]  Laboratory of Nanostructured Materials Physics, Institute of Molecule and Crystal Physics of Ufa Federal Research Centre RAS, Prospekt Oktyabrya 151, 450075 Ufa, Russia
[3]  "Nanotech" Engineering Center, Ufa State Aviation Technical University, 12 K. Marx Str, 450008 Ufa, Russia; v.astanin@gmail.com
[4]  Electron and Ion Beam Lab, Department of Solid State Electronics, Physics Faculty, Saint Petersburg State University, 7/9 Universitetskaya nab., 199034 Saint Petersburg, Russia; ubyivovk@gmail.com
[5]  Frank Laboratory of Neutron Physics, Joint Institute for Nuclear Research, 141980 Dubna, Russia; Islamov@nf.jinr.ru
[6]  School of Materials Science and Engineering, Nanjing University of Science and Technology, Nanjing 210094, China; jtwang@njust.edu.cn
*   Correspondence: dimagun@mail.ru

**Abstract:** Vit105 ($Zr_{52.5}Cu_{17.9}Ni_{14.6}Al_{10}Ti_5$ at. %) bulk metallic glass samples were processed by high-pressure torsion and accumulative high-pressure torsion. By DSC, XRD and SANS methods it was shown that accumulative high-pressure torsion allows for achieving high real strains and leads to an increase in the free volume and significant transformation of the structure. Minor crystallization was detected after high-pressure torsion processing.

**Keywords:** bulk metallic glass; free volume; high pressure torsion

## 1. Introduction

High strength, large elastic strain limit, good corrosion resistance and other unique properties provide a great potential for the commercial application of bulk metallic glasses (BMGs). The structure of bulk metallic glasses determines their mechanical properties [1–3]. In particular, BMG strength and yield stress are higher then their crystalline counterparts [1].

Unfortunately, BMGs exhibit low tensile ductility. The deformation of BMG occurs via the formation of shear bands (SB), and BMGs fracture along the very first shear band [1,2,4]. It is known that the formation of nanoscale heterogeneities—clusters and nanocrystals—can lead to an increase in plasticity. For example, as shown in [5–9], preliminary deformation by rolling or compression enables the increase in ductility of amorphous alloys. The main idea of these routes is to form in BMGs nanometer-sized heterogeneities or to obtain amorphous structures consisting of nanoclusters [8]. A preliminary deformation by compression or cold rolling enables increasing the ductility of amorphous alloys [5–7]. This is attributed to the fact that the shear bands produced by the preliminary deformation, distributed in the bulk of a BMG, are the source generating multiple shear bands during subsequent loading. Deformation becomes more uniform and ductility increases. HPT is an efficient

way of producing a high strain, and, as a result, the formation of a high density of shear bands in BMG. In crystalline materials, HPT processing results in the formation of "ultrafine" grains or a nanocrystalline structural state [10,11].

Numerous studies have shown the effect of HPT on the structure and properties of amorphous alloys [11–27]—in particular, reviews have been recently published [28,29]. HPT deformation leads to partial nanocrystallization in some amorphous alloys [28–30]. In other amorphous alloys, nanocrystallization during HPT was not observed; however, in these alloys during HPT, a complex change in the structure occurs, a free volume increases and the heterogeneities are formed, and this all leads to a change in alloy properties.

Free volume $\Delta V$ is an important characteristic of amorphous alloys [1–4]. In particular, the presence of free volume provides some ductility in amorphous materials (during the deformation by compression, rolling, three-point bending) [2,3]. Most studies demonstrate that HPT processing results in the growth of free volume in the amorphous phase [13,30]. Amorphous samples after HPT have a high density of shear bands. The samples of HPT-processed amorphous alloys contain many shear bands [31,32]. The SB structure differs from the structure of the amorphous matrix, also in terms of density [4]. Therefore, the structure of amorphous alloys after HPT contains two phases—amorphous matrix and shear bands. The increase in free volume can take place both due to the accumulation of shear bands in the structure and due to the changes in the amorphous matrix [4].

The first coordination sphere radius ($R_1$) can be obtained by X-ray diffraction (XRD). According to [33–37], X-ray diffraction (XRD) allows you to evaluate the change in $\Delta V$ amorphous alloys. Studies have shown that HPT leads to an increase in $\Delta V$ [33,37]. Changes in $\Delta V$ from XRD data may have a significant error. However, in [33], the increase in the free volume content in HPT processing BMG $Zr_{62}Cu_{22}Al_{10}Fe_5Dy_1$ was revealed both by XRD and by direct density measurements. Still, it is necessary to further investigate the evaluation of $\Delta V$ in BMGs after HPT processing, using both XRD and differential scanning calorimetry (DSC), since the data on the changes in free volume can be obtained from the data on the relaxation energy of amorphous alloys [13].

Shear strain, $\gamma$, introduced during HPT processing, is calculated using the following equation:

$$\gamma = \frac{2\pi nR}{h},$$ (1)

where $R$ is the distance from center of the disk-shaped sample, $n$ is the number of revolutions and $h$ is the disk thickness [10].

However, as shown in [12,31,38–40], the real strain in the disk subjected to HPT is lower in comparison with that estimated from (1). Due to this, the authors used a new method: "accumulative high pressure torsion" (ACC HPT) [38], which allows higher strains on amorphous alloys. In the present paper, the studies into the effect of "accumulative HPT" processing on the Zr-BMG are continued, using a range of experimental research methods and different processing regimes of conventional HPT and accumulative HPT.

## 2. Materials and Methods

The Vit105 ($Zr_{52.5}Cu_{17.9}Ni_{14.6}Al_{10}Ti_5$ at. %) BMG plates with a size of 60 mm × 10 mm × 2 mm were used in this study. The plates were produced by squeeze casting into a massive copper mold with a melt cooling rate of $10^3$ K/s.

HPT processing was carried under a pressure $P = 6$ GPa on ⌀10 mm anvils with a 0.3 mm depth groove, with a rotation speed of 1 rpm, at room temperature (RT). Samples with a diameter of 10 mm and a thickness of 1 mm were cut out from the as-cast BMG plates for HPT processing.

Part of the BMG disks were subjected to conventional HPT for $n = 5$, $n = 10$ and $n = 30$ full revolutions. Some disks were subjected to the accumulative HPT (ACC HPT) procedure, which was reported in [38]. During the ACC HPT procedure, at the first stage, a disk-shaped sample with a thickness of $h = 1$ mm and ⌀10 mm were subjected to HPT for two anvil revolutions, $n = 2$,

under a pressure of 6 GPa. As a result, a disk with $h \approx 0.5$ mm and $\varnothing \approx 10$ mm was obtained. At the second stage, the HPT disk produced at the first stage was cut into four segments, and these segments were stacked on the HPT-anvils on top of each other. As a result, a stack with a total height $h \approx 2$ mm was obtained. A pressure of 6 GPa was imposed on this stack, HPT processing for $n = 2$ revolutions was performed again. The cycles of "accumulative HPT for $n = 2$" were repeated three times. At the last stage, the cycle was repeated, but with 4 turns of the anvils, and the result was a monolithic sample. The total number of revolutions of the anvils during ACC HPT processing was $n_{\Sigma} = 10$. At each stage, the shear strain produced by HPT processing for $n = 2$ was summed with the strain produced by the compression of segment stack, and the total strain was the sum of strains produced at all four stages. Very approximate estimates show that the total strain produced during "accumulative HPT for $n = 10$", due to the summation of compressive and torsional HPT strains, was about 2–3 times larger than that during conventional HPT for $n = 10$.

The XRD analysis of the samples was performed on the Rigaku Ultima IV (Rigaku Corporation, Tokyo, Japan) diffractometer using $CuK_{\alpha}$ monochromatized radiation in the reflected beam, and a parabolic graphite monochromator was used. The diffractometer X-ray tube operated at a voltage and current of 40 kV and 40 mA, respectively. The Bragg–Brentano scheme of goniometer was used. Measurements were performed at a field size of $1 \times 1.5$ mm$^2$, located in the area that is 2.5 mm from the disc center. The relationship between the intensity of reflected X-ray quantum's and diffraction angle $2\theta$ was within $10°$ to $80°$ in increments of $0.02°$ and with 6 s of exposure time per point. XRD data were analyzed using PHILIPS ProFit software. Netzsch DSC 204 F1 Phoenix calorimeter (NETZSCH-Gerätebau GmbH, Selb, Germany) was used for DSC testing, and Netzsch Proteus software (ver. 4.8.5) was used for DSC analysis; the heating temperature was 500 °C, which is higher than the crystallization temperature for Vit105. The heating rate was 20 K/min. In order to determine more precisely the relaxation energy, the DSC tests were conducted in the following manner: first, heating to a temperature slightly above $T_g$ (420 °C), then cooling to 80 °C; then a second heating to a temperature slightly above the temperature of the crystallization finish ($T_f = 500$ °C), cooling; then a third heating to the temperature $T_f$. The relaxation energy $H_{relax}$ was found from the difference in the enthalpies of the first and second heatings in a region of 80–420 °C. The crystallization energy, $H_x$, was found from the difference in the enthalpies of the second and third heatings in the crystallization region. Characteristic points of the curves were determined as described in ISO 11357-1:2009.

The structure of both as-cast and HPT-processed specimens was investigated on a ZEISS Libra 200FE (Carl Zeiss Microscopy GmbH, Oberkochen, Germany) transmission electron microscope (TEM) at an acceleration voltage of 200 kV, in both high-resolution TEM (HRTEM) and dark field TEM modes. The samples for TEM were prepared by the focused ion beam (FIB) technique using an ZEISS Auriga Crossbeam SEM-FIB (Carl Zeiss Microscopy GmbH, Oberkochen, Germany), and finished using Gatan PIPS (Pleasanton, CA, United States) ion polishing system. The selected area electron diffraction (SAED) data were collected from an area of 80 nm in diameter.

The hardness of specimens was measured using a microhardness tester Emco-Test Durascan 50 (EMCO-TEST Prüfmaschinen GmbH, Kuchl, Austria) with a Vickers-type indenter. The static load was 1 N (100 g) and the dwell time of loading was 10 s. Five indentations were conducted at each measurement point (the center of the HPT-processed disk-specimen and the edge of the HPT-processed disk-specimen) for each specimen.

Studies by small-angle neutron scattering were performed at the Joint Institute for Nuclear Research, Dubna, Russia. Small-angle neutron scattering (SANS) experiments were performed on the YuMO time-of-flight spectrometer at high flux pulse IBR-2 reactor [41,42]. YuMO is an instrument with two movable detector systems placed at sample-to-detectors distances of 5.28 and 13.4 m, resulting in a $Q$ range of 0.006–0.5 Å$^{-1}$, where $Q = \dfrac{4\pi}{\lambda} \sin\left(\dfrac{\theta}{2}\right)$ and $\theta$ is the scattering angle.

The measured neutron scattering spectra were corrected for the transmission and thickness of the sample, background scattering on the film substrate and on the vanadium reference sample using SAS software [43], yielding a neutron scattering intensity in absolute units of cm$^{-1}$.

## 3. Results and Discussion

As revealed by transmission electron microscopy, the as-cast BMG is amorphous (Figure 1a). The BMG after HPT processing is also predominantly amorphous (Figure 1b). However, the appearance of the microstructure slightly changes. Fragmentation appears on rings on SAED images, taken from an area of 60 nm in diameter (Figure 1b, inset), while HRTEM (Figure 1b) photos reveal some fringes that might mean some kind of ordering in the structure.

The appearance of the dark field also changes slightly (Figure 1e). Individual regions of about 5 nm in size are visible in the dark field. Apparently, some strain-induced nanocrystallization begins in Vit105 during HPT processing. However, the areas that are visible in the dark field in HPT-treated Vit105, if they are crystallites, are very small—less than 5 nm—and have an internal structure, because they consist of many luminous points, which is not typical for equilibrium nanocrystals. The luminous inhomogeneous regions observed in the dark field are more likely to indicate the formation/growth of amorphous clusters. The structure and composition of clusters/crystallites have not yet been identified due to their small amount. It should be noted that in other areas of the same sample, the diffraction pattern shows the amorphous state of the material, and strain-induced nanocrystallization is not observed (Figure 1c).

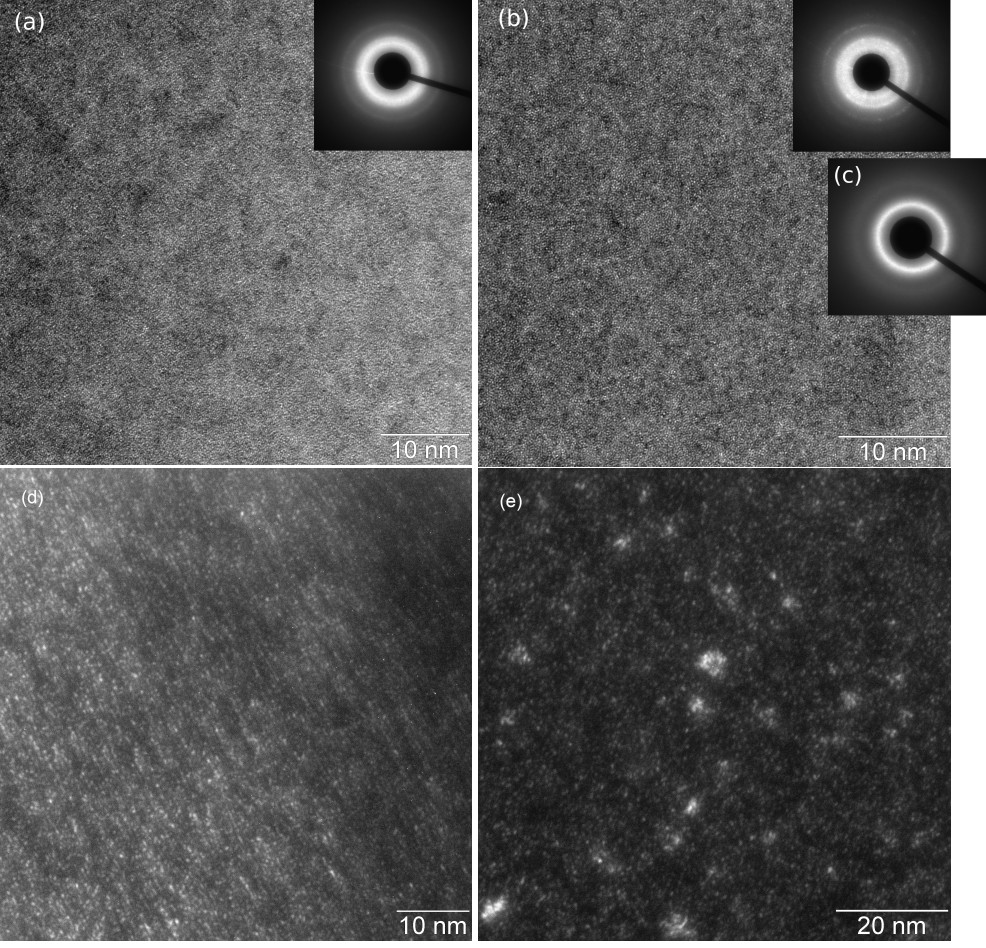

**Figure 1.** Microstructure of the Vit105 BMG: (**a**) as-cast (HRTEM, SAED); (**b**) subjected to HPT for *n* = 30 (HRTEM, SAED); (**c**) subjected to HPT for *n* = 30, SAED in an area without strain-induced nanocrystallization; (**d**) as-cast, TEM, dark field, (**e**) HPT *n* = 30, TEM, dark field.

Strain-induced nanocrystallization has also been observed earlier for some other amorphous alloys [15–17,19,24], but in Zr-based BMG, it is apparently less distinct and was not noted in many works [21]. Cluster formation along with nanocrystallization in HPT amorphous alloy was observed

earlier in [44,45]. It can be assumed that the formation of clusters in HPT amorphous alloy is the initial stage of nanocrystallization. Shear bands were not observed in any of the HPT-processed Vit105 samples by TEM, which is related to the small contrast of shear bands in Zr-based BMG; shear bands could not be detected by TEM in many other works either [13,14,21]. Perhaps, more detailed and precise TEM studies will enable revealing the features in the structure of the HPT-processed Vit105, but at the moment the interpretation of the fine features of structure requires further experimental investigations.

XRD shows that the structure of the original alloy, and after HPT, is amorphous (Figure 2). Crystalline peaks, even if observed in HPT-processed samples, are very small, and their intensity indicates an extremely low content of the crystalline phase. Here we note that, according to [46], even 10% content of the crystalline phase can give a clearly visible peak in the diffractogram of the amorphic alloy. Therefore, in this case, SPD-induced nanocrystallization does not play a decisive role in the transformation of the structure, unlike, for example, an amorphous NdFeB alloy and an amorphous Al-based alloy, where SPD-induced nanocrystallization plays a crucial role [15–17,24]. The position of the maximum first amorphous halo of the Vit105 BMG shifts towards the lowest angles after HPT processing (Table 1). According to [34,35], the radius of the first coordination sphere $R_1$ and the relative changes in free volume $\Delta V$ were found from the position of the maximum first amorphous halo. The values of $R_1$ and $\Delta V$ obtained in accordance with the mentioned procedure are undoubtedly very approximate, since the position of the averaged maximum of the first amorphous halo can change, due the decomposition of the amorphous phase into two amorphous phases of slightly differing compositions, or due to partial nanocrystallization, which is not taken into consideration in this procedure. However, this procedure enables evaluating the variation intensity of the amorphous structure resulting from external action, using conventional XRD.

According to these calculations, HPT processing for $n = 5$ leads to a $\approx$2% increase in the $\Delta V$ and also leads to an increase in the values of full width at half maximum ($FWHM$) by 4.2%. The increase in $FWHM$ resulting from HPT processing could be related both to an increase in free volume [36] and a possible separation of the initially homogeneous amorphous phase into two or more constituents, slightly different in composition [35,47]. In the XRD pattern, the diffusion peaks from them usually merge and form a common halo [47]. Consequently, the parameters $R_1$ and $\Delta V$, found from the position of the common halo's gravity center, together with $\Delta FWHM$ can be used only tentatively to estimate the degree of the structural transformations in the amorphous alloy during HPT processing. Conventional HPT processing for $n = 10$ also leads to a slight increase in the values of $FWHM$—up to 5%—which is supposed to indicate a further growth of the non-homogeneity of Vit105. According to the calculations, conventional HPT processing for $n = 10$ leads to a 0.3% decrease in $\Delta V$. However, the negative value of $\Delta V$ (i.e., a decrease in free volume) could be the result of a large error in determining $R_1$ and $\Delta V$ from the position of the averaged maximum of the first amorphous halo, additionally broadened after HPT processing. HPT processing for $n = 30$ leads to an increase in the values of $FWHM$ by 3%, and to a $\approx$4% increase in $\Delta V$. Such a large obtained value of $\Delta V$ could also be the result of a large error in finding $R_1$ and $\Delta V$ from the position of the averaged maximum first amorphous halo, but it confirms the tendency for $\Delta V$ to grow as a result of HPT processing. ACC HPT processing for $n_\Sigma = 10$ leads to an increase in $\Delta V$ by $\approx$2%, and the increase in the $FWHM$ is greater—up to 17% (Table 1), which indicates a more intensive structural transformation of Vit105 during ACC HPT processing.

According to the DSC data, the relaxation energy $H_{relax}$, as a result of HPT processing, also steadily increases with an increase in $n$—from $\approx$1.5 J/g for the as-cast state to 11 J/g after HPT processing for $n = 30$ (Table 2)—which indicates the growth enthalpy and the free volume of amorphous alloys [13,14]. This is correlated with the XRD data. As a result of ACC HPT processing, the increase in the $H_{relax}$ of Vit105 is greater, up to 29 J/g, which indicates a more intensive structural transformation and growth of the $\Delta V$ of the amorphous alloy during ACC HPT. The heat, which is released during relaxation and determined using DSC as the difference between the first and the second heatings, appears at 150 °C for the ACC HPT-processed samples, which is noticeably lower than in the case of the as-cast

Vit105 and Vit105 after conventional HPT. It can be assumed that ACC HPT processing leads to the appearance of some types of defects/structural elements in the amorphous phase, relaxing at a relatively low temperature. Such defects in the amorphous phase were reported earlier, in particular, in [48]. Relaxation energy and free volume are connected by the relationship $H_{relax} = A \cdot \Delta V$, where $A$ is a certain constant [37]. According to [37], the constant $A$ for Zr-based amorphous alloys can be estimated as $A = 9$ J/g. Therefore, $\Delta V_H = H_{relax}/A$. The free volume, $\Delta V_{H_{rel}}$, calculated from $H_{relax}$ confirms the growth of free volume of the BMG during HPT processing, and especially after ACC HPT processing. It should be noted that, in the accepted calculation procedure, the "free volume" $\Delta V_{H_{rel}}$ is determined with respect to a fully relaxed state, while $\Delta V$, determined from the XRD data, is relative to the as-cast state of the BMG. The data on the variation of $\Delta V_H$ obtained from relaxation energy apparently correspond better to the actual picture than the values of $\Delta V$, obtained from the XRD data.

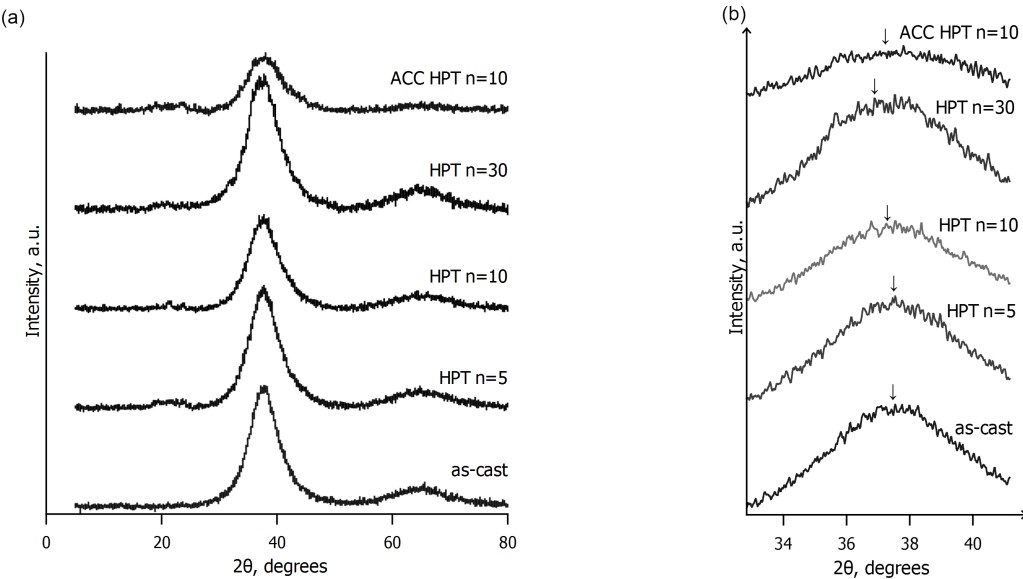

**Figure 2.** XRD results for the Vit105 BMG: as-cast; after HPT for $n = 5$, $n = 10$, $n = 30$; after accumulative HPT for $n_\Sigma = 10$: (**a**) overview; (**b**) enlarged halo area, the arrow indicates the position of the halo gravity center.

**Table 1.** XRD data on the structure of Vit105 in various conditions: BMG Vit105 after HPT via different regimes and accumulative HPT for $n_\Sigma = 10$.

| Condition | $2\theta$, ° | $R_1$, Å | $FWHM$, ° | $\Delta V$, % | $\Delta FWHM$, % | $\Delta V_H$, % |
|---|---|---|---|---|---|---|
| as-cast Vit105 | 37.46 | 2.951 | 6.40 | | | 0.2 |
| HPT $n = 5$ | 37.50 | 2.969 | 6.48 | 1.8 | 4 | 0.6 |
| HPT $n = 10$ | 37.29 | 2.948 | 6.73 | −0.3 | 5 | 0.8 |
| ACC HPT $n_\Sigma = 10$ | 37.22 | 2.969 | 7.48 | 1.8 | 17 | 3.2 |
| HPT $n = 30$ | 36.90 | 2.995 | 6.59 | 4.5 | 3 | 1.2 |

$\Delta V$ is the free volume value estimated from the $R_1$ shift on XRD, estimated relatively to as-cast state; $\Delta V_H$ is the free volume value estimated from relaxation energy obtained by DSC method relative to fully relaxed state.

The crystallization of Vit105 takes place in two stages, as indicated by two peaks in the DSC diagram (Figure 3). This was observed earlier in similar BMGs [49]. The temperature, $T_x$, of the Vit105 BMG does not change very much as a result of HPT processing. The view of crystallization DSC peak changes considerably as a result of ACC HPT processing. The peaks of the two crystallization stages merge, the temperature of crystallization onset slightly increases, by 10 °C, which is apparently related to a change in the first stage of crystallization. As a rule, HPT deformation on various BMGs leads to a decrease in the temperature of crystallization onset [32,50,51]. However, in the case of two-stage crystallization, which takes place in this alloy, HPT deformation, on the contrary, increases it

(Figure 3). An increase in the crystallization temperature as a result of HPT was observed earlier in [14]. To analyze the changes in the crystallization processes as a result of HPT and ACC HPT processing, further experimental studies are required.

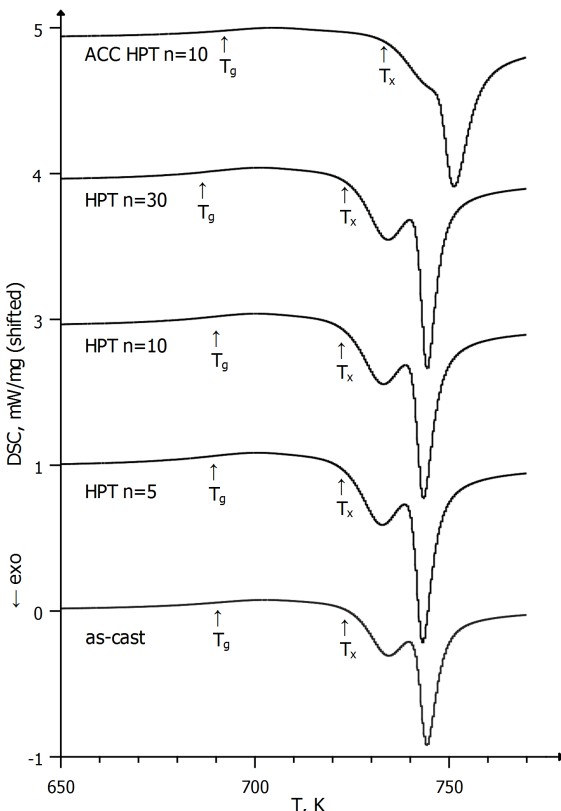

**Figure 3.** DSC curves for Vit105 in the as-cast state and after HPT via different regimes.

**Table 2.** DSC data for the Vit105 BMG after HPT via different regimes HPT and accumulative (ACC) HPT.

| Condition | $H_{relax}$, J/g | $\Delta V_H$, % | $T_g$, °C | $\Delta C_p$, J/g·K | $T_x$, °C | $T_{p1}$, °C | $T_{p2}$, °C | $H_x$, J/g |
|---|---|---|---|---|---|---|---|---|
| as-cast | 1.5 | 0.2 | 418 | 0.38 | 451 | 461 | 471 | 45 |
| HPT $n = 5$ | 5 | 0.6 | 416 | 0.49 | 450 | 460 | 470 | 58 |
| HPT $n = 10$ | 7 | 0.8 | 417 | 0.48 | 450 | 460 | 470 | 58 |
| ACC HPT $n = 10$ | 29 | 3.2 | 419 | 0.35 | 460 | - | 478 | 55 |
| HPT $n = 30$ | 11 | 1.2 | 414 | 0.47 | 451 | 461 | 471 | 56 |

$H_{relax}$—relaxation energy; $\Delta V_H$—free volume value estimated from the $H_{relax}$; $T_g$—glass transition temperature; $\Delta C_p$—heat capacity change during glass transition; $T_x$—crystallization temperature; $T_{p1}$, $T_{p2}$—crystallization peaks; $H_x$—crystallization enthalpy.

**Table 3.** Microhardness of the Vit105 samples, processed by HPT via different regimes.

| Condition | Center | Edge |
|---|---|---|
| As-cast | 525 | |
| HPT $n = 5$ | 500 | 497 |
| HPT $n = 10$ | 506 | 497 |
| ACC HPT $n_\Sigma = 10$ | 508 | 480 |

Microhardness studies show that, as a result of HPT processing, the *HV* of Vit105 declines relative to the as-cast state (Table 3). The decrease in the *HV* of BMGs as a result of HPT processing was also observed earlier in other works [21,52]. The decrease in the *HV* of BMGs as a result of HPT processing

is related to an increase in free volume. The decrease in *HV* is especially intense at the edge of the HPT-processed samples, which is attributed to the larger strain at the edge of the HPT-processed samples [11]. After ACC HPT processing, microhardness at the edge of the Vit105 samples decreases even further. It is also known that the value of *HV* at different points of an HPT-processed sample is influenced in a complex manner by the pattern of microstress distribution [39], which can explain, in particular, the inconsistent variation in *HV* in the center of the HPT-processed sample.

Low resolution of the SANS technique cannot give the detailed structure at the atomic level but is useful to describe structural features typically at distances from 1 to 100 nm [53]. In the case of disordered systems, such as metallic glasses, in the absence of strong correlations between inhomogeneities (fluctuations in the scattering density), the scattered intensity can follow the power law of *Q*:

$$I(Q) = A \cdot Q^{-\alpha} + B, \tag{2}$$

where *A* and *B* are constants, and *B* is the background. It was shown [54] that for objects with complex branched surfaces (surface fractal of dimension from $2 \leq Ds \leq 3$) $3 \leq \alpha \leq 4$, scattering can be given by:

$$I(Q) = A \cdot Q^{-(1-Ds)} + B, \tag{3}$$

$$A = \pi N_0 \Delta\rho^2 \Gamma(5 - Ds) \sin\left(\frac{\pi(Ds - 1)}{2}\right),$$

where $N_0$ is the measure of the fractal surface and $\Delta\rho$ is the difference between coherent scattering density of the inner part of amorphous cluster and its boundary (surface) layer, which have the different composition of elements or less density packing. The scattering densities of elements $Zr_{52.5}Cu_{17.9}Ni_{14.6}Al_{10}Ti_5$ were calculated using [55] and are given in Table 4.

**Table 4.** The scattering densities of elements included in $Zr_{52.5}Cu_{17.9}Ni_{14.6}Al_{10}Ti_5$.

| Element | $Zr_{52.5}Cu_{17.9}Ni_{14.6}Al_{10}Ti_5$ | Zr | Cu | Ni | Al | Ti |
|---|---|---|---|---|---|---|
| $\rho, 10^{10}, cm^{-2}$ | 3.717 | 3.077 | 6.540 | 9.414 | 2.078 | −1.910 |

SANS data of Vit105 in the as-cast state, after HPT $n = 10$ and after ACC HPT $n_\Sigma = 10$ are presented in Figure 4. The results of fits with (2) are given in Table 5. $Q_{max}$ corresponds to the intersection of the model curve with the background from the sample and approximately gives an estimate of the minimal size inhomogeneity of the scattering density in the sample, as $d \sim 2\pi/Q_{max}$.

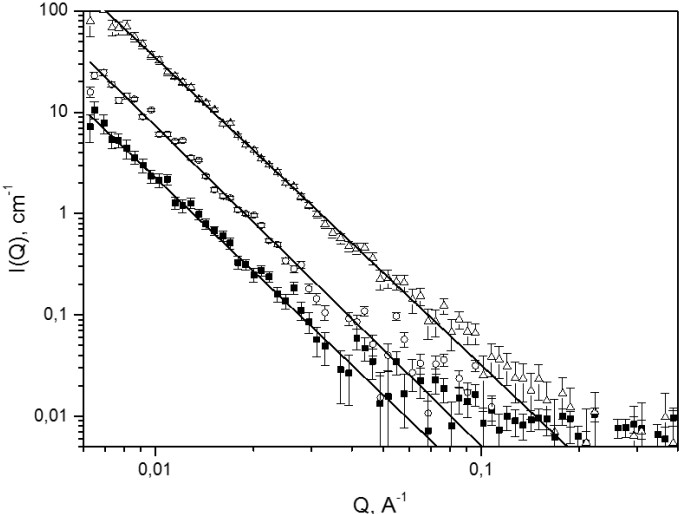

**Figure 4.** Small-angle neutron scattering from Vit105: as-cast (■); after HPT $n = 10$ (○); after ACC HPT $n = 10$ (△).

At this stage, it is impossible to determine the size distribution and volume fraction of clusters, since their composition and difference in elements are unknown, but only the general elemental composition of BMG is available. Since the scattering intensity is proportional to the square of the difference between the scattering densities at the boundary and inside the cluster (contrast factor), to the volume and concentration of clusters, then in the formula for determining the particle size distribution, one can obtain a large difference in scattering either due to a change in the volume concentration of clusters or due to a combination various elements forming phases at the interface. In this regard, the analysis of the Porod's invariants admits great arbitrariness and does not allow us, at the moment, to estimate the size distribution of clusters.

**Table 5.** The results of experimental data fit with (2).

| Condition | A | $\alpha$ | B, cm$^{-1}$ | $Q_{max}$, Å$^{-1}$ |
|---|---|---|---|---|
| as-cast | $1.69 \times 10^{-6}$ | $3.1 \pm 0.1$ | 0.005 | 0.073 |
| HPT $n = 10$ | $3.83 \times 10^{-6}$ | $3.1 \pm 0.1$ | 0.005 | 0.1 |
| ACC HPT $n = 10$ | $29.9 \times 10^{-6}$ | $3.1 \pm 0.1$ | 0.005 | 0.19 |

Figure 4 shows comparative changes in scattering behavior before and after different HPT modes. All analyzed states are characterized by the same $Q^{-3.1}$ scattering law-scattering from a highly branched surface, not necessarily closed. This approach was used to analyze the minimum size of clusters/regions with sections existing in the amorphous phase. At the same time, Porod's approach to the analysis of scattering curves is based on the $Q^{-4}$ scattering law from a smooth surface closing the cluster. The analysis of the scattering curves within the framework of the $Q^{-3.1}$ scattering law showed that, in all considered states, the morphology of the structures (the hierarchy of the sizes and distribution of clusters) remains identical.

SANS data analysis shows that, in the as-cast state, the amorphous phase consists of some structural inhomogenities (conditionally clusters), with a minimal size approximately beginning from $d > 2\pi/Q_{max} = 8.3$ nm.

It should be noted that these clusters have not yet been able to be revealed on existing equipment using the TEM method, as well as in [49] on same Vit105 BMG.

After HPT $n = 10$, the minimal structural inhomogenities are observed as beginning from 6.3 nm and after ACC HPT $n = 10$ from 3.3 nm. It is interesting to note that the scattering for the as-cast state and after treatment procedure follow to the same power law with the exponent $\alpha = 3.1$ (so-called surface fractals, objects with complex rough surfaces).

The structures of these samples are similar on large sizes. More intensive treatment gives changes in the cluster interface at smaller scales, while the surface area between the clusters increases.

The significant increase in scattering intensity (see Table 5) can be qualitatively explained by two factors. The first is a change in the contrast of the scattering density between the amorphous cluster and the boundary layer due to the release of weakly coupled elements to the cluster surface. For example, the release of *Ti*, *Al*, *Ni* to the surface can increase the contrast several times (Table 4). Accordingly, the contrast in the second power can give an increase in intensity by an order of magnitude. Perhaps this is the result of the emergence on the interface of clusters of elements loosely coupled in the amorphous phase. This separation is not significant, occurs at the nanoscale and is not revealed by other structural research methods, only by SANS.

The second factor is the increase in surface area, the $N_0$ parameter in (3) as a result of cluster size reduction after HPT processing. A decrease in the size of the amorphous cluster in an amorphous alloy leads to an increase in the interface between amorphous clusters and, as a result, to an increase in free volume, which is also observed by DSC. Unfortunately, it is impossible to unambiguously determine the role of the first or second parameter due to the lack of additional information. Moreover, ACC HPT leads to stronger grinding of SANS-registered clusters than ordinary HPT and, accordingly,

to a larger increase in free volume, which is confirmed by other data (DSC, XRD, HV). The results of the parameters are presented in Table 5.

We shall note that the SANS results on some changes in the cluster structure as a result of HPT are confirmed by the previously reported TEM data. In general, the structural transformations observed in HPT Vit105 are quite complex. On the one hand, there is evidence of the beginning of some strain-induced nanocrystallization and cluster growth (TEM), which is rather a relaxation process and should lead to a decrease in the free volume. On the other hand, XRD and DSC indicate an increase in the free volume and internal energy of the amorphous phase. An increase in the free volume simultaneously with the formation of clusters and nanocrystals as a result of HPT should lead to the multiplication of shear bands under loading and an increase in the plasticity of the amorphous alloy, as was observed earlier [14,56]. Thus, as a result of HPT, the structure of the amorphous phase becomes more complex and heterogeneous, as evidenced by SANS data, but the nature of these changes requires additional research and analysis.

## 4. Conclusions

Influence of HPT and accumulative HPT on the microstructure of the Zr-based BMG Vit105 ($Zr_{52.5}Cu_{17.9}Ni_{14.6}Al_{10}Ti_5$ at. %) was investigated.

Strain-induced nanocrystallization was observed after HPT processing in some areas of Vit105. However, the crystallite size appears to be very small, in the observed region of about 5 nm, and there are few crystals.

According to XRD, the position of the maximum first amorphous halo of the Vit105 BMG shifts towards lower angles and *FWHM* increases after HPT processing, which supposedly means an increase in $R_1$ and ($\Delta V$). ACC HPT processing for $n_\Sigma = 10$ leads to an increase in $\Delta V$ and the increase in the *FWHM* is greater than conventional HPT processing, which indicates a more intensive structural transformation of Vit105 during ACC HPT processing.

According to the DSC data, the relaxation energy $H_{relax}$ as a result of HPT processing also steadily increases with an increase in $n$ from $\approx$1.5 J/g for the as-cast state to 11 J/g after HPT processing for $n = 30$, which indicates growth of the enthalpy and the free volume of amorphous alloys. As a result of ACC HPT processing, the increase in the $H_{relax}$ of Vit105 is larger, up to 29 J/g, which indicates a more intensive structural transformation and growth of the $\Delta V$ of the amorphous alloy during ACC HPT.

Microhardness studies show that, as a result of HPT processing, the HV of Vit105 decreases relative to the as-cast state. After treatment with ACC HPT, the microhardness decreases even more noticeably, which also indicates a greater increase in $\Delta V$.

According to a small-angle neutron scattering study, the amorphous cluster is reduced in size after HPT processing. In this case, ACC HPT leads to a more significant decrease in the sizes of clusters, as obtained by SANS, than conventional HPT. A decrease in the size of the amorphous cluster leads to an increase in the interface between amorphous clusters and, as a result, to an increase in free volume, which is also confirmed by other data (DSC, XRD, HV).

Perhaps the change in the SANS pattern is also associated with some strain-induced nanocrystallization as a result of HPT, but the physics of these changes requires additional research and analysis.

**Author Contributions:** Conceptualization, D.G.; methodology, D.G. and A.I.; validation, A.C.; formal analysis, V.A. and A.C.; investigation, V.S., E.U. and A.I.; resources, J.T.W.; data curation, V.A.; writing—original draft preparation, D.G.; writing—review and editing, V.A.; visualization, V.A.; supervision, J.T.W.; project administration, D.G.; funding acquisition, D.G. and J.T.W. All authors have read and agreed to the published version of the manuscript.

**Funding:** This work was supported by the Ministry of Education of the Russian Federation, project 0838-2020-0006 "Fundamental study of new principles for the creation of promising electromechanical energy converters with characteristics above the world level, with increased efficiency and minimum specific indicators, using new highly efficient electrotechnical materials"; the Natural Science Foundation of China, grant No. 51520105001; HPT processing funded by the Russian Foundation for Basic Research, grant No. 20-08-00497, and part of the experimental study-by the Russian Foundation for Basic Research, Project No. 19-58-80018 (BRICS_t).

**Acknowledgments:** TEM studies were performed using the facility of IRC Nanotechnology of Science Park St-Petersburg State University. Part of the work was carried out using the equipment of the "Nanotech" Engineering Center (http://nanotech.ugatu.ac.ru).

**Conflicts of Interest:** The authors declare no conflict of interest.

## Abbreviations

The following abbreviations are used in this manuscript:

| | |
|---|---|
| ACC HPT | Accumulative high pressure torsion |
| BMG | Bulk metallic glass |
| DSC | Differential scanning calorimetry |
| FIB | Focused ion beam |
| FWHM | Full width at half maximum |
| HPT | High-pressure torsion |
| HRTEM | High-resolution transmission electron microscopy |
| RT | Room temperature |
| SAED | Selected area electron diffraction |
| SANS | Small-angle neutron scattering |
| TEM | Transmission electron microscopy |
| XRD | X-Ray diffraction |

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
