# Peer review of "Influence of High-Pressure Torsion and Accumulative High-Pressure Torsion on Microstructure and Properties of Zr-Based Bulk Metallic Glass Vit105"

_metals, doi:10.3390/met10111433_

Round 1
Reviewer 1 Report
It is a very interesting work, Worth to be published.
A good and well documented bibliography is provided. Although the techniques used for the investigation of amorphous structures are not the most suitable ones, the authors present, nevertheless, convincing hints and trends from their experimental results due to a rigorous and efficient treatment of the data they could get. The presented manuscript reveals altogether a good work which should, however, be confirmed by additional experiments in the future.
Despite of this good description, the manuscript needs the following modifications before the it can be accepted for publication in the present state of the study:
1) The authors have performed SANS experiments at the Dubna facility. Why ND experiments were not added?
As the authors write themselves, "XRD data may have a significant error." As the authors surely are aware of, standard XRD does not give enough statistics because of the reduced irradiated area. Thus, for the investigation of amorphous structure neutron diffraction is the most appropriate technique as it has been reported extensively in the literature by groups all over the world. Standard XRD is suitable for crystalline structure not for amorphous ones.
Indeed, the statistics and resolution of Fig2 is very poor. Thus, although the authors are showing trends with the results presented in Table1, it cannot be concluded from Fig2 that the state of the investigated samples is fully amorphous. This is even confirmed by the authors themselves in the comments of the TEM results that some regions are amoprhous and some not.
Accordingly, te text should be rewritten all over the manuscript in order to take into account this limitation.
2) The discussion on pages 8 and 9 is the key part. Thus, more should be extracted from the SANS data.
Fig4 is not convincing. Has a POROD study been performed on the obtained data? If yes, the obtained distribution of the scattering particles should be shown and discussed in detailed correlation with the DSC and diffraction results.
Reviewer 2 Report
The submitted paper discusses about influence of high-pressure torsion and accumulative high-pressure torsion on the microstructure and properties of the Zr-based bulk metallic glass Vit105. The authors used high-pressure torsion and accumulative high-pressure torsion to process vit105 bulk metallic glass specimens. The experiments showed that high real strains appeared after accumulative high-pressure torsion and this leads to an increase in the free volume and significant transformation of the structure. Moreover, the authors observed strain-induced nanocrystallization after HPT processing in some areas of Vit105. However, in my opinion, the paper has been poorly organized and there is no suitable characterization on tested samples. There is no SEM, EBSD or TEM characterization on high-pressure torsion and accumulative high-pressure torsion tested specimens. The provided data is not enough for a research paper. Therefore, the paper cannot be accepted in the current format by metals.
Round 2
Reviewer 2 Report
The authors convinced me that they they did not get any significant result from SEM and EBSD. They implied that the SEM did not present any separation of the amorphous phase by elements on the scales that our equipment allows to obtain. Moreover, the EBSD technique showed no decomposition. They explained that based on the XRD analysis on tested specimens, the intensity of peaks was very low. Therefor, the paper can be accepted for publication in the current format.